# Emergence of Ceftazidime- and Avibactam-Resistant *Klebsiella pneumoniae* Carbapenemase-Producing *Pseudomonas aeruginosa* in China

Yiwei Zhu,[a,b,c] Jie Chen,[a,b,c*] Han Shen,[d] Zhongju Chen,[e] Qi-wen Yang,[f,g] Jin Zhu,[h] Xi Li,[i] Qing Yang,[j] Feng Zhao,[k] Jingshu Ji,[k] Heng Cai,[a,b,c] Yue Li,[a,b,c] Linghong Zhang,[a,b,c] Sebastian Leptihn,[a] Xiaoting Hua,[a,b,c] Yunsong Yu[a,b,c]

[a]Department of Infectious Diseases, Sir Run Run Shaw Hospital, Zhejiang University School of Medicine, Hangzhou, China

[b]Key Laboratory of Microbial Technology and Bioinformatics of Zhejiang Province, Hangzhou, China

[c]Regional Medical Center for National Institute of Respiratory Diseases, Sir Run Run Shaw Hospital, Zhejiang University School of Medicine, Hangzhou, China

[d]Department of Laboratory Medicine, Nanjing Drum Tower Hospital, Nanjing University Medical School, Nanjing, China

[e]Department of Laboratory Medicine, Tongji Hospital, Tongji Medical College, Huazhong University of Science and Technology, Wuhan, China

[f]Department of Laboratory Medicine, Chinese Academy of Medical Sciences, Beijing, China

[g]Beijing Key Laboratory for Mechanisms Research and Precision Diagnosis of Invasive Fungal Diseases, Peking Union Medical College Hospital, Peking Union Medical College, Beijing, China

[h]Department of Clinical Laboratory, Quzhou People's Hospital, Affiliated Quzhou Hospital of Wenzhou Medical University, Quzhou, China

[i]Centre of Laboratory Medicine, Zhejiang Provincial People's Hospital, People's Hospital of Hangzhou Medical College, Hangzhou, China

[j]State Key Laboratory for Diagnosis and Treatment of Infectious Diseases, Collaborative Innovation Center for Diagnosis and Treatment of Infectious Diseases, The First Affiliated Hospital, College of Medicine, Zhejiang University, Hangzhou, China

[k]Department of Clinical Laboratory, Sir Run Run Shaw Hospital, Zhejiang University School of Medicine, Hangzhou, China

**ABSTRACT** *Klebsiella pneumoniae* carbapenemase (KPC)-producing *Pseudomonas aeruginosa* (KPC-PA) has been reported sporadically. However, epidemiological and antimicrobial susceptibility data specific for KPC-PA are lacking. We collected 374 carbapenem-resistant *P. aeruginosa* (CRPA) isolates from seven hospitals in China from June 2016 to February 2019 and identified the $bla_{KPC-2}$ gene in 40.4% ($n = 151/374$) of the isolates. Approximately one-half of all KPC-PA isolates ($n = 76/151$; 50.3%) were resistant to ceftazidime-avibactam (CAZ-AVI). Combining Kraken2 taxonomy identification and Nanopore sequencing, we identified eight plasmid types, five of which carried $bla_{KPC-2}$, and 13 combination patterns of these plasmid types. In addition, we identified IS*26*-ΔTn*6296* and Tn*1403*-like–ΔTn*6296* as the two mobile genetic elements that mediated $bla_{KPC-2}$ transmission. $bla_{KPC-2}$ plasmid curing in 28 strains restored CAZ-AVI susceptibility, suggesting that $bla_{KPC-2}$ was the mediator of CAZ-AVI resistance. Furthermore, the $bla_{KPC-2}$ copy number was found to correlate with KPC expression and, therefore, CAZ-AVI resistance. Taken together, our results suggest that KPC-PA is becoming a clinical threat and that using CAZ-AVI to treat this specific pathogen should be done with caution.

**IMPORTANCE** Previous research has reported several cases of KPC-PA strains and three KPC-encoding *P. aeruginosa* plasmid types in China. However, the prevalence and clinical significance of KPC-PA are not available. In addition, the susceptibility of the strains to CAZ-AVI remains unknown. Samples in this study were collected from seven tertiary hospitals prior to CAZ-AVI clinical approval in China. Therefore, our results represent a retrospective study establishing the baseline efficacy of the novel $\beta$-lactam/$\beta$-lactamase combination agent for treating KPC-PA infections. The observed correlation between the $bla_{KPC}$ copy number and CAZ-AVI resistance suggests that close monitoring of the susceptibility of the strain during treatment is required. It would also be beneficial to screen for the $bla_{KPC}$ gene in CRPA strains for antimicrobial surveillance purposes.

Address correspondence to Xiaoting Hua, xiaotinghua@zju.edu.cn, or Yunsong Yu, yvys119@zju.edu.cn.

*Present address: Jie Chen, Department of Laboratory, Ningbo Medical Center Lihuili Hospital, Ningbo, China.

**KEYWORDS** carbapenem-resistant *Pseudomonas aeruginosa*, $bla_{KPC-2}$, ceftazidime-avibactam, *Pseudomonas aeruginosa*

Pseudomonas aeruginosa belongs to the most common opportunistic pathogens. Increasing numbers of clinical isolates show resistance to antibiotics, including carbapenems. Although carbapenems are still among the first-line therapeutics for infections caused by multidrug-resistant (MDR) *P. aeruginosa* strains, carbapenem-resistant *P. aeruginosa* (CRPA) strains are increasingly being observed in the clinic. CRPA belongs to the pathogens listed by the World Health Organization (WHO) that are considered of high relevance for human health and for which new antibiotics or clinical strategies are urgently needed (1).

Approved by the U.S. Food and Drug Administration (FDA) in 2015, ceftazidime-avibactam (CAZ-AVI), a novel $\beta$-lactam/$\beta$-lactamase inhibitor (BLBLI) combination agent, has been deployed in the clinic for complicated intra-abdominal infections and hospital-acquired pneumonia caused by multidrug-resistant *Enterobacteriaceae* and *P. aeruginosa* (2). The inhibition spectrum of avibactam includes the *Klebsiella pneumoniae* carbapenemase (KPC) family (3). According to International Network for Optimal Resistance Monitoring (INFORM) global surveillance programs (4–6), CAZ-AVI susceptibility rates are between 84 and 90% to CRPA strains that do not express metallo-$\beta$-lactamase (MBL). However, currently, no study has detailed the CAZ-AVI susceptibility of KPC-producing *P. aeruginosa* (KPC-PA) as a single group and investigated resistance mechanisms, possibly due to its relatively lower prevalence than its counterpart in carbapenem-resistant *Enterobacteriaceae* (CRE) (7), although the last decade has witnessed an increasing number of $bla_{KPC}$ genes detected in clinical *P. aeruginosa* isolates (8–13).

The antibiotic resistance gene (ARG) $bla_{KPC}$ is most commonly plasmid borne (14–17). The NCBI GenBank database lists 17 complete KPC-encoding *P. aeruginosa* plasmids, among which 3 plasmid types have been reported in China (see Table S1 in the supplemental material). The first type was identified recently in East China and has not been assigned to any IncP type yet (15, 16). The second type belongs to a megaplasmid family associated with multiple ARGs, spreading widely around the world (17). The third type is an IncP-6 plasmid with a novel $bla_{KPC}$-adjacent region (18). However, most of these studies were case reports and focused on the description of $bla_{KPC}$ and its adjacent sequences. Therefore, the distribution of KPC-encoding *P. aeruginosa* plasmids remains unclear.

In this study, we analyzed the prevalence of KPC-PA strains from seven hospitals in China from June 2016 to February 2019 and tested *in vitro* antimicrobial susceptibility. We used Illumina and Nanopore sequencing to elucidate the molecular epidemiology and genetic characteristics of the KPC-PA strains. Our data revealed a high rate of resistance (50.3%) to CAZ-AVI of KPC-PA in China. In addition, we deciphered the plasmidome of KPA-PA and identified five KPC-carrying plasmid types. Furthermore, we found that the $bla_{KPC-2}$ copy number correlated with CAZ-AVI resistance.

## RESULTS

**Geographic distribution, antimicrobial susceptibility, and clinical data.** A total of 374 CRPA clinical isolates were collected from seven hospitals in China (Sir Run Run Shaw Hospital [SRRSH], *n* = 86 [23.0%]; First Affiliated Hospital of Zhejiang University [FAHZU], *n* = 71 [19.0%]; Wuhan Tongji Hospital [WTJH], *n* = 50 [13.4%]; Peking Union Medical College Hospital [PUMCH], *n* = 50 [13.4%]; Nanjing Drum Tower Hospital [NDTH], *n* = 44 [11.8%]; Provincial People's Hospital of Zhejiang [ZPPH], *n* = 38 [10.2%]; Quzhou People's Hospital [QZPH], *n* = 35 [9.4%]). A total of 151 (40.4%) strains were $bla_{KPC}$ positive based on PCR screens. All $bla_{KPC}$ genes were $bla_{KPC-2}$. $bla_{KPC-2}$ genes were detected in CRPA isolates from all hospitals with the exception of PUMCH. The percentage of $bla_{KPC-2}$-positive strains of CRPA varied among hospitals from 11.9% (NDTH) to 92.1% (ZPPH) (Fig. 1A). Antimicrobial susceptibility tests of the 151 KPC-PA isolates

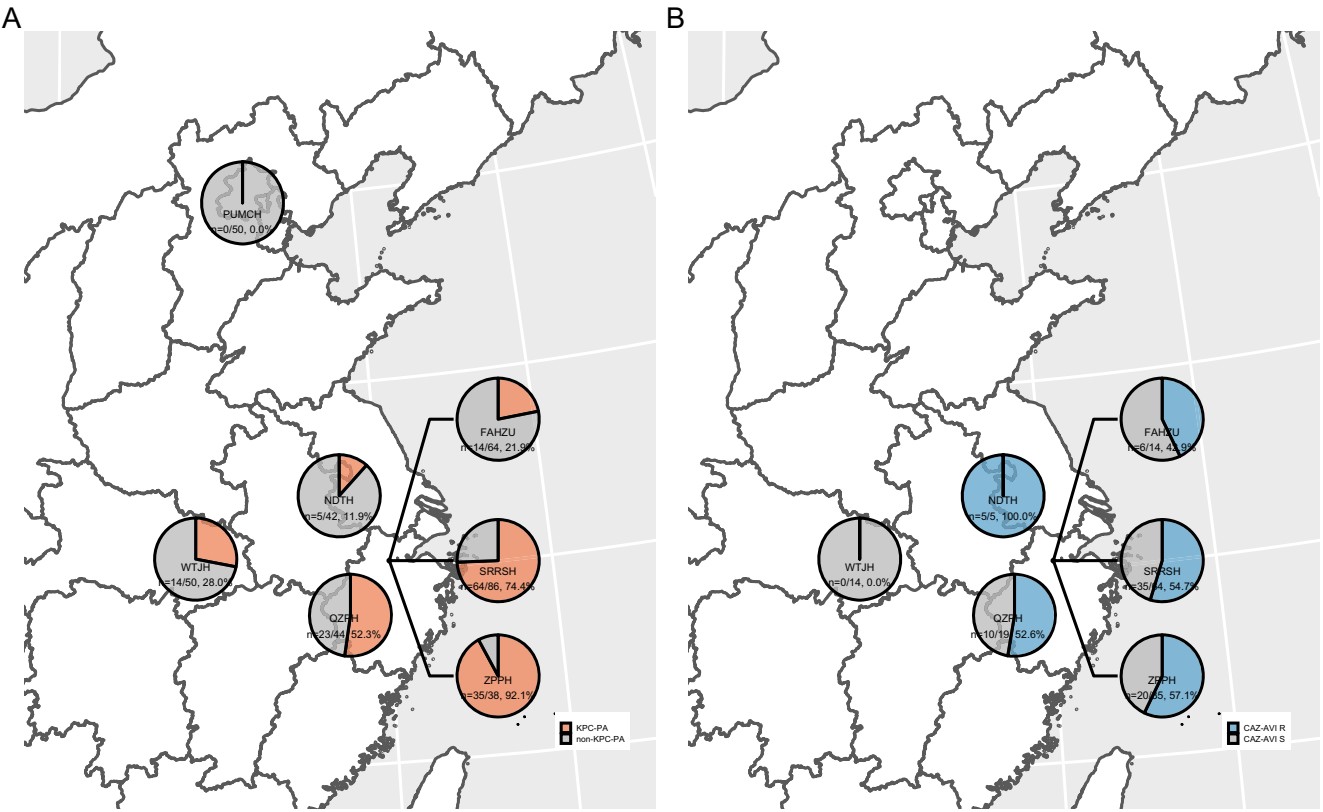

**FIG 1** Geographic distribution of strains in this study. (A) Percentages represent the ratios of KPC-PA isolates of CRPA strains from each hospital. (B) Percentages represent the ratios of CAZ-AVI-resistant *P. aeruginosa* isolates of KPC-PA strains from each hospital. PUMCH, Peking Union Medical College Hospital; WTJH, Wuhan Tongji Hospital; NDTH, Nanjing Drum Tower Hospital; FAHZU, First Affiliated Hospital of Zhejiang University; SRRSH, Sir Run Run Shaw Hospital; ZPPH, Provincial People's Hospital of Zhejiang; QZPH, Quzhou People's Hospital; R, resistant; S, susceptible. The maps were drawn using the R package mapchina (https://github.com/xmc811/mapchina), and the data source was derived from https://www.openstreetmap.org.

revealed high-level (>90%) resistance to all β-lactams except CAZ-AVI and fluoroquinolones (Table 1). Most isolates ($n = 141/151$; 93.4%) met the criteria of difficult-to-treat resistance (19). Half of the isolates ($n = 76/151$; 50.3%) were resistant to CAZ-AVI. The CAZ-AVI resistance rate of KPC-PA in each hospital varied substantially (Fig. 1B). The aminoglycoside resistance rates were low, ranging from 4.0% to 8.6%. No strain was resistant to colistin. Three strains from NDTH were extensively drug resistant and susceptible only to colistin. Complete results are shown in Table S2 in the supplemental material.

Most of the CRPA samples were collected from intensive care units (ICUs) ($n = 115$; 30.7%) and general surgery departments ($n = 85$; 22.7%). There was no statistically significant difference in the distributions of KPC-PA and non-KPC-PA isolates among clinical departments. Of note, KPC-PA was strongly associated with urine and abdominal drainage samples (Fisher exact odds ratios [ORs] [95% confidence intervals {CIs}] were 7.48 [3.25 to 19.35] and 3.29 [1.63 to 6.87] [adjusted $P < 0.0001$]) (Table S3). The high carriage rate of $bla_{KPC}$ genes in *P. aeruginosa* from these samples endorsed the role of novel class A carbapenemase inhibitors in the treatment of intra-abdominal infection and urinary tract infection as the indications for CAZ-AVI.

**Molecular epidemiology.** Making use of Illumina sequencing data available for 151 KPC-PA strains, we found that multilocus sequence typing (MLST) indicated three main KPC-PA sequence types (STs), ST463 ($n = 107$; 70.9%), ST485 ($n = 14$; 9.3%), and ST1212 ($n = 12$; 7.9%), dominating in different geographic regions (Fig. 2). ST463 was mainly found in cities in East China (Nanjing [$n = 5/5$; 100%], Hangzhou [$n = 85/113$; 75.2%], and Quzhou [$n = 17/19$; 89.5%]), while ST485 ($n = 14/14$; 100%) is the main KPC-PA ST of Wuhan in Central China.

**TABLE 1** *In vitro* antimicrobial susceptibility tests of KPC-producing *P. aeruginosa* strains from 6 hospitals in China[a]

| Antibiotic | MIC (mg/liter) | | | % resistance | CLSI cutoff (mg/liter) |
|---|---|---|---|---|---|
| | MIC$_{50}$ | MIC$_{90}$ | MIC range | | |
| TZP | >256/4 | >256/4 | 16/4 to >256/4 | 99.34 | 128/4 |
| CAZ | 128 | 256 | 32 to >256 | 100.00 | 32 |
| FEP | >256 | >256 | 16 to >256 | 99.34 | 32 |
| IPM | >128 | >128 | 32 to >128 | 100.00 | 8 |
| MEM | >128 | >128 | 16 to >128 | 100.00 | 8 |
| ATM | >128 | >128 | 32 to >128 | 100.00 | 32 |
| CAZ-AVI | 16/4 | 32/4 | 2/4 to >64/4 | 50.33 | 16/4 |
| AK | 4 | 16 | 1 to >64 | 3.97 | 64 |
| CN | 4 | 8 | 0.25 to >64 | 8.61 | 16 |
| TOB | 1 | 2 | 0.25 to >64 | 3.97 | 16 |
| CIP | >16 | >16 | 0.12 to >16 | 94.04 | 2 |
| LEV | >32 | >32 | 0.5 to >32 | 93.38 | 4 |
| CO | 0.5 | 0.5 | <0.03 to 2 | 0.00 | 4 |

[a]TZP, piperacillin-tazobactam; CAZ, ceftazidime; FEP, cefepime; IPM, imipenem; MEM, meropenem; ATM, aztreonam; CAZ-AVI, ceftazidime-avibactam; AK, amikacin; CN, gentamicin; TOB, tobramycin; CIP, ciprofloxacin; LEV, levofloxacin; CO, colistin.

**Kraken2-based plasmid contig classification of the *P. aeruginosa* plasmidome.** Eight plasmid types were identified in the CRPA strains. According to their combination in each strain (plasmid pattern), we assigned all the strains to 13 groups named patterns A to M (Fig. 2 and Table 2). Pattern A (*n* = 100; 66.2%) and pattern B (*n* = 19; 12.6%) were the major plasmid patterns, representing strains harboring only type I and II plasmids, respectively. Twenty-one isolates of five plasmid patterns contained more than one plasmid type individually. An association of some plasmid types with STs was observed. The type I plasmids were mainly found in ST463 and ST1212 strains. The type II plasmids were most commonly encountered in ST485 and ST463 strains, and the type VIII plasmids were exclusively carried by ST1212 strains. We selected representatives from each plasmid pattern for Nanopore long-read sequencing. Five KPC-encoding plasmid types were obtained (Fig. 3; a detailed description of each plasmid type can be found in the supplemental information text and Fig. S1 to S5 at https://doi.org/10.6084/m9.figshare.14802648.v6).

Analysis of each of the plasmid types revealed several common features of these plasmids. First, despite the diversity of plasmid types, the core *bla*$_{KPC-2}$ genetic platform ISKpn27-*bla*$_{KPC-2}$-ISKpn6 remained identical in almost all samples. There were two prototypes of *bla*$_{KPC-2}$-associated mobile genetic elements, IS26-ΔTn6296 and Tn1403-like–ΔTn6296 (see Fig. S1B and Fig. S6 at https://doi.org/10.6084/m9.figshare.14802648.v6), both of which belonged to the Tn21 subfamily of the Tn3 family of transposons (20). The identical core *bla*$_{KPC-2}$ platform indicated that the promoter region variation probably did not contribute to KPC overexpression and CAZ-AVI resistance. Novel promoter regions were detected only in strains FAHZU40 and NDTH9845 (see Fig. S3C and Fig. S7 at https://doi.org/10.6084/m9.figshare.14802648.v6). Second, the *bla*$_{KPC-2}$-adjacent region beyond the core platform varied by multiple IS26-mediated inversion or duplication events, amplifying the *bla*$_{KPC-2}$ gene dosage. It could be supported by the comparison of plasmids pSRRSH1002-KPC and pSRRSH1408-KPC (see Fig. S1C at https://doi.org/10.6084/m9.figshare.14802648.v6). In addition, aside from plasmids, the integration of *bla*$_{KPC-2}$-associated mobile genetic elements (MGEs) into the chromosome also participated in reshaping the chromosome structure. For instance, on the strain NDTH10366 chromosome, the three tandem repeats of the *bla*$_{KPC-2}$-associated MGE array promoted two large chromosome segment translocations (see Fig. S8 at https://doi.org/10.6084/m9.figshare.14802648.v6).

**CRISPR/Cas9-based plasmid curing facilitates the identification of *bla*$_{KPC-2}$-associated MGE locations and their impact on CAZ-AVI susceptibility.** As described above, *bla*$_{KPC-2}$ was located in a transposon like Tn6296, which made it difficult to identify

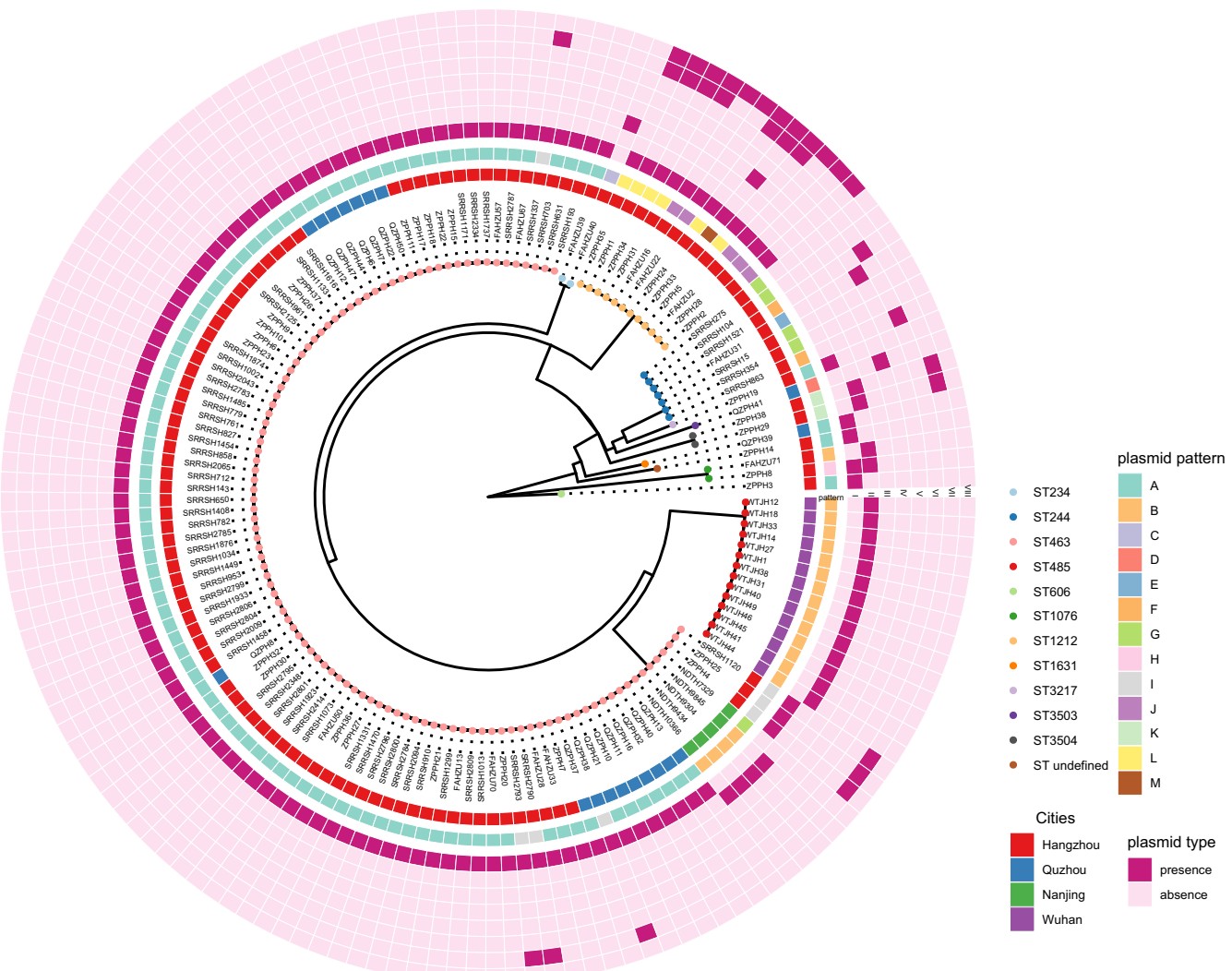

**FIG 2** Core-genome phylogenetic tree and plasmid patterns. The innermost layer is a maximum likelihood phylogenetic tree of KPC-PA strains. The colors of the tip labels indicate sequence types (ST). The second ring indicates cities where the strains were isolated. The third ring represents the classification of the plasmid composition in each strain. The outer heat map represents plasmid types resident in each strain.

its location unambiguously based on short-read mapping. Although we selected representatives of each plasmid pattern to perform Nanopore long-read sequencing, it was still uncertain to infer the $bla_{KPC-2}$ location in the remaining strains. To address this problem, we implemented a CRISPR/Cas9-based plasmid-curing system. As an example, we took the type I plasmid. It was reasonable to conclude that the $bla_{KPC-2}$ gene was located only on the type I plasmid if the gene was not detected by PCR after the plasmid had been removed; otherwise, the gene might additionally be located on the chromosome or other coexisting plasmids. With this high-throughput method, we successfully cured the type I plasmids in 28 strains. The $bla_{KPC-2}$ gene was still detected in four strains after type I plasmid curing, indicating that it had transposed to other replicons. The successful curing of $bla_{KPC}$-carrying plasmids was further demonstrated by the restoration of CAZ-AVI susceptibility (Table 3). Our results supported the conclusion that in most scenarios, the $bla_{KPC-2}$ genes were located in only one replicon and that transposition to other sites was less frequent. After curing type I plasmids, all the isolates ($n = 28$) turned susceptible to CAZ-AVI, including the strains harboring $bla_{KPC-2}$ at other locations (chromosome and/or other plasmids). This indicated that multiple $bla_{KPC-2}$ copies were the major mechanisms for CAZ-AVI resistance in *P. aeruginosa*. Reducing the copy number and not necessarily eliminating all $bla_{KPC-2}$ copies might also mitigate CAZ-AVI resistance.

**TABLE 2** Plasmid patterns and types defined by Kraken2 taxonomy classification

| Plasmid pattern | Plasmid type(s) | No. of strains | Representative strain |
|---|---|---|---|
| A | I | 100 | SRRSH1002 |
| B | II | 19 | WTJH12 |
| C | III | 1 | FAHZU40 |
| D | IV | 1 | QZPH41 |
| E | V | 1 | FAHZU31 |
| F | VI | 2 | SRRSH1521 |
| G | No plasmid | 5 | SRRSH15 |
| H | I, II | 1 | ZPPH8 |
| I | I, VII | 7 | SRRSH2790 |
| J | I, VIII | 5 | ZPPH2 |
| K | II, VII | 2 | ZPPH29 |
| L | I, VII, VIII | 6 | ZPPH1 |
| M | I, IV, VII, VIII | 1 | ZPPH33 |

**$bla_{KPC-2}$ copy number correlated with CAZ-AVI resistance.** CAZ-AVI is a comparably new $\beta$-lactam/$\beta$-lactamase inhibitor (BLBLI) playing an important role in treating infections caused by KPC-producing organisms. In our samples, almost all KPC-PA strains exhibited high-level resistance to carbapenems (>128 mg/liter), while the susceptibility to CAZ-AVI varied in a large range (2/4 to 512/4 mg/liter). Based on the results described above, we further investigated the correlation between $bla_{KPC-2}$ copy numbers and CAZ-AVI MIC values. Two strains containing MBLs (NDTH9845 and NDTH10366) were excluded since avibactam had no inhibitory effect on MBLs. We found a clear correlation of the $bla_{KPC-2}$ gene copy numbers with the CAZ-AVI MIC values (Spearman rank correlation [$\rho$] = 0.491; $P < 0.0001$). Statistically significant differences in the $bla_{KPC-2}$ gene copy numbers were observed among CAZ-AVI MIC groups (2/4 to 32/4 mg/liter) (Fig. 4A). However, when high levels of resistance ($\geq$64/4 mg/liter) were observed, the relationship was less obvious, possibly due to the comparably few strains exhibiting high-level resistance to CAZ-AVI ($n = 6$).

We further quantified KPC expression in 10 selected strains. We found that the $bla_{KPC-2}$ expression level was significantly higher in the CAZ-AVI-resistant group than in the susceptible group ($P < 0.001$ by a Wilcoxon rank sum test) (Fig. 4B). The $bla_{KPC-2}$ expression level correlated with its gene copy number (Spearman $\rho$ = 0.58; $P < 0.001$) (Fig. 4C) and also the CAZ-AVI MIC (Spearman $\rho$ = 0.67; $P < 0.0001$) (Fig. 4D). This supported a gene dosage effect with multiple gene copies increasing the level of gene expression.

**Impact of other AMR mechanisms on CAZ-AVI resistance.** Besides $bla_{KPC-2}$, other chromosomal ARGs had also been linked to CAZ-AVI resistance, including those encoding the *Pseudomonas*-derived cephalosporinase (PDC), the efflux pump MexAB-OprM, and penicillin-binding protein 3 (PBP3) (2). Overexpression of PDC and its variants that could contribute to CAZ-AVI resistance was not detected. The overexpression of the efflux pump MexAB-OprM and the F533L mutation in PBP3 might contribute to CAZ-AVI resistance (see Fig. S9 at https://doi.org/10.6084/m9.figshare.14802648.v6; see also Table S2 in the supplemental material). However, restoring CAZ-AVI susceptibility after curing KPC-encoding plasmids indicated that these ARG effects were minor.

## DISCUSSION

In this study, we focused on a specific subpopulation of CRPA: KPC-PA. At the point of time when the bacteria were isolated, the KPC-PA strains showed moderate susceptibility (50.3%) to CAZ-AVI. The MICs of approximately 70% of strains were at the breakpoint margin (8/4 to 16/4 mg/liter). Deploying Kraken2 taxonomy classification and Nanopore long-read sequencing, we were able to decipher the plasmidome of KPC-PA and identify five KPC-encoding plasmid types. We found that the $bla_{KPC-2}$ copy number variation was caused by mobile genetic elements, in particular IS*26*-mediated

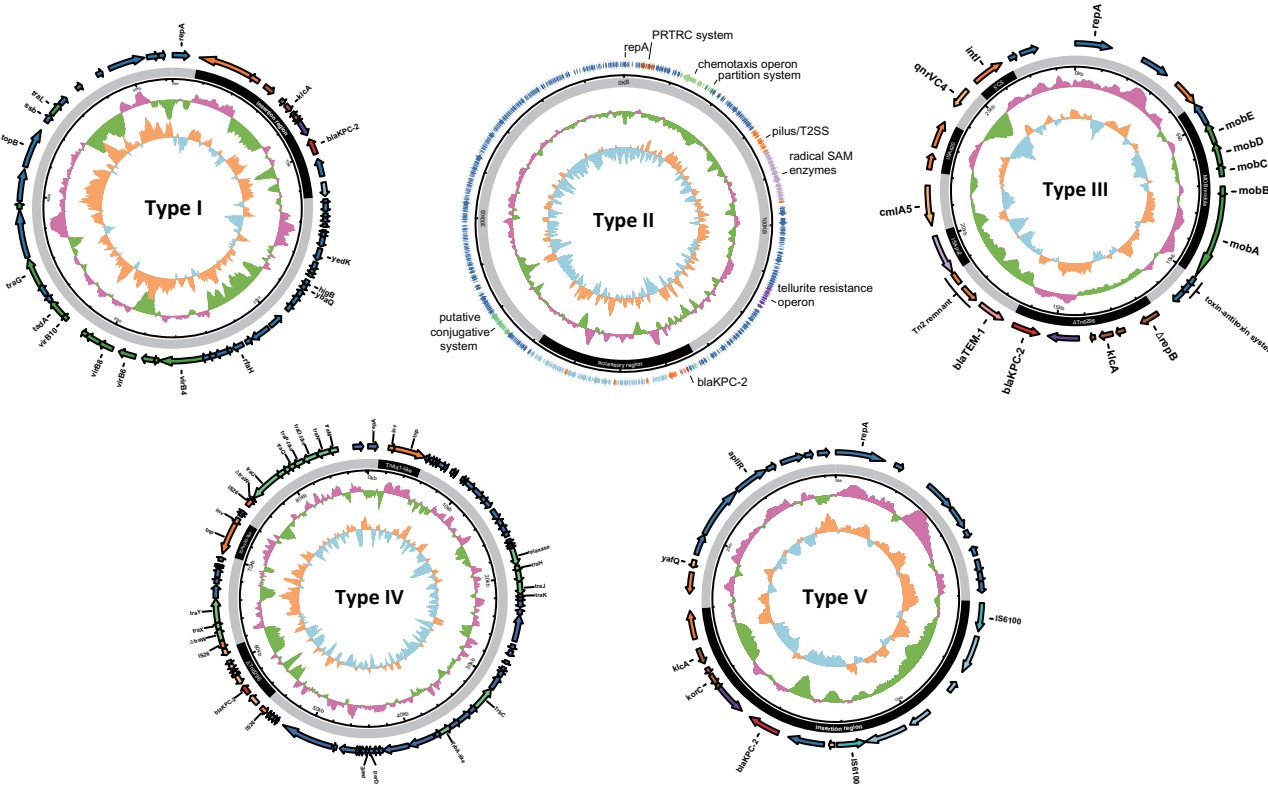

**FIG 3** Five KPC-encoding plasmids identified in this study. For each plasmid plot, the innermost ring represents the GC skew, where sky blue means a skew to A and T and orange means a skew to G and C. The second ring from the innermost ring represents the GC content, where light green represents a percent GC content higher than the median and dark pink represents a percent GC content lower than the median (sliding window of 500 bp). The third ring shows the backbone (gray) and insertion region (black) of the plasmid. The outmost ring shows genes on representatives of each plasmid type (GenBank accession number): type I, pZPPH1-KPC (CP077990); type II, pWTJH12-KPC (CP064404); type III, pFAHZU40-KPC (CP078008); type IV, pQZPH41-KPC (CP064400); type V, pFAHZU31-KPC (CP078010). 5′CS, 5′ common sequence; PRTRC, ParB-related, ThiF-related cassette; T2SS, type II secretion system; SAM, S-adenosylmethionine.

transposition. Combining sequencing depth calculation, plasmid curing, and gene expression measurement, we revealed that the $bla_{KPC-2}$ copy number correlated with CAZ-AVI resistance.

Previous epidemiological surveillance studies had focused on CRPA. One of the largest studies performed in China reported that rates of resistance of CRPA to CAZ-AVI were 34.3% in 2017 (21) and 35.7% in 2018 (22). The strains investigated in our study were collected around the same period. Therefore, we inferred that KPC-PA exhibited a higher CAZ-AVI resistance rate than the overall CRPA population. No screening was performed regarding the $bla_{KPC-2}$ gene in these surveillance studies, which reflected the unawareness of which important role this ARG played in *P. aeruginosa*.

The origin of the $bla_{KPC-2}$ gene in *P. aeruginosa* is probably an interspecies transmission event between *P. aeruginosa* and *K. pneumoniae*, as the $bla_{KPC-2}$-associated MGEs

**TABLE 3** Antimicrobial susceptibilities before and after type I plasmid curing[a]

| | Before curing | | After curing | |
|---|---|---|---|---|
| Antibiotic | MIC$_{50}$ (mg/liter) | MIC$_{90}$ (mg/liter) | MIC$_{50}$ (mg/liter) | MIC$_{90}$ (mg/liter) |
| IPM | >128 | >128 | 8 | >128 |
| MEM | >128 | >128 | 64 | >128 |
| CAZ | 256 | 256 | 2 | 32 |
| CAZ-AVI | 16/4 | 64/4 | 2/4 | 4/4 |

[a]IPM, imipenem; MEM, meropenem; CAZ, ceftazidime; CAZ-AVI, ceftazidime-avibactam.

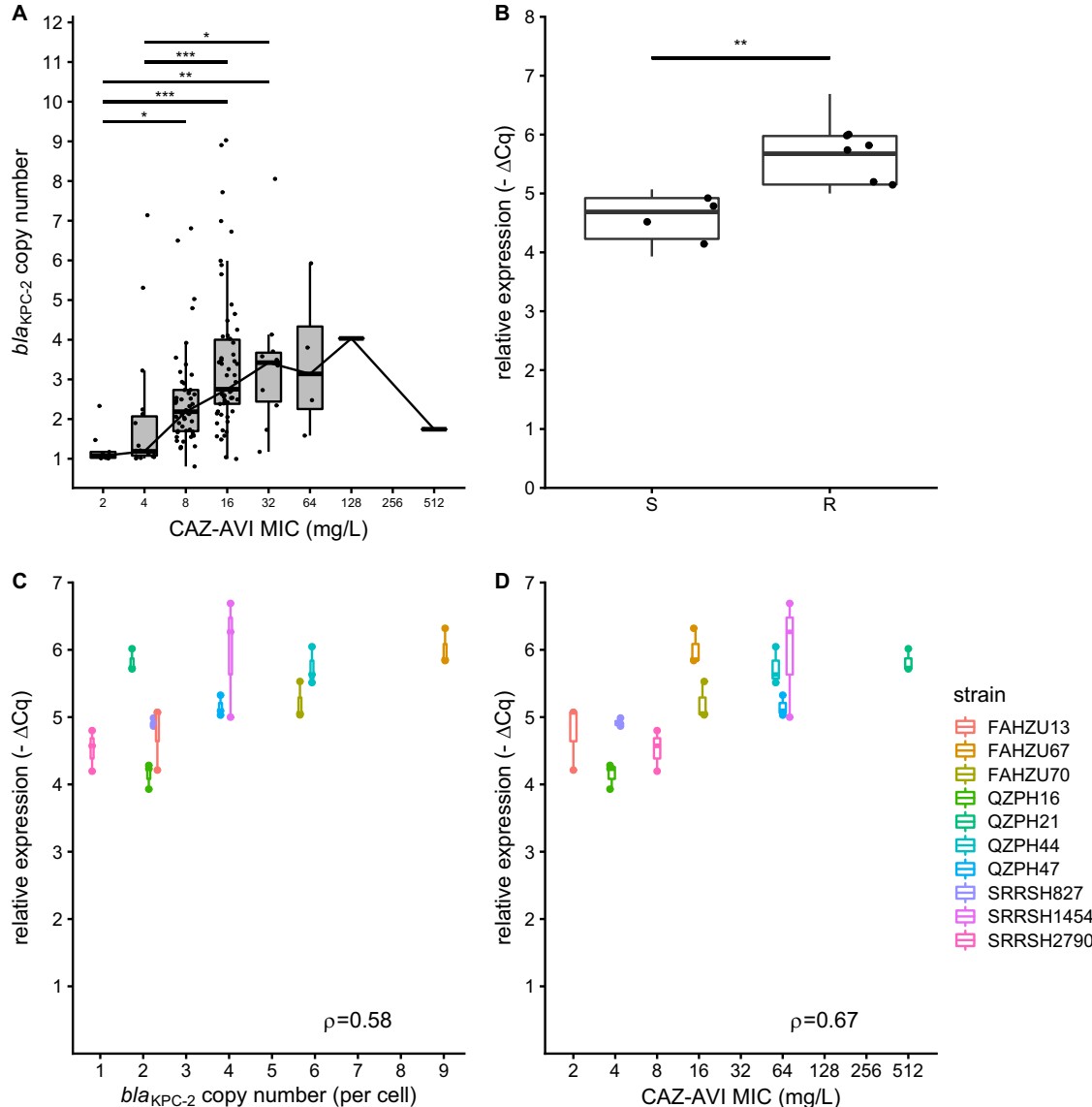

**FIG 4** Correlation among $bla_{KPC-2}$ copy numbers, expression levels, and CAZ-AVI MICs. (A) $bla_{KPC-2}$ gene copy number comparison among strains ($n = 149$) in different CAZ-AVI MIC groups. *, **, and *** represent adjusted $P$ values of <0.05, <0.01, and <0.001, respectively, as determined by a Dunn test. (B) $bla_{KPC-2}$ transcription comparison between susceptible ($n = 4$) and resistant ($n = 6$) strains (Wilcoxon rank sum exact test $W = 60$; $P < 0.01$). (C) Correlation between $bla_{KPC-2}$ copy numbers and KPC transcription levels ($n = 10$) (Spearman $\rho = 0.58$; $P < 0.001$). (D) Correlation between CAZ-AVI MICs and KPC transcription levels ($n = 10$) (Spearman $\rho = 0.67$; $P < 0.0001$). Box plots indicate data from biological triplicates for each strain. Cq, quantification cycle.

were nearly identical (23). A previous study (24) indicated that 65% of carbapenem-resistant *K. pneumoniae* isolates in China carried the $bla_{KPC-2}$ gene, which made it the most prevalent carbapenemase in this species and might have been horizontally transferred to *P. aeruginosa*. It should also be noted that most KPC-encoding plasmid types have not yet been found in *Enterobacteriaceae*. It is possible that the transfer of enterobacterial plasmids to *P. aeruginosa*, in which they cannot replicate and be maintained, still gives the $bla_{KPC-2}$-associated MGEs on them an opportunity to transpose onto resident plasmids or the chromosome (20). The emergence of the predominant ST463 clone is worrisome. It is possible that capturing the KPC-encoding plasmid allows it to survive in clinical environments where the antibiotic pressure is high (25). Such specific associations between plasmid types and bacterial clones have been observed in previous studies (26). It leaves for future investigation why the specific combination of a type I plasmid and an ST463 strain is so successful in the clinic.

Our study found a correlation between $bla_{KPC-2}$ copy number variation and CAZ-AVI resistance. Previous studies had demonstrated that $\beta$-lactamase gene amplification correlated with susceptibility to $\beta$-lactam/$\beta$-lactamase inhibitor combination agents (27–29). Our results showing that insertion sequence (IS)-mediated gene amplification contributes to CAZ-AVI MIC elevations support these conclusions.

The CAZ-AVI resistance mechanisms have been explored in previous studies. As for KPC-producing organisms, the most common mechanisms were substitutions in KPC, especially in the $\Omega$-loop (2). In our study, KPC variants were not identified. CAZ-AVI-resistant KPC variants in *P. aeruginosa* were sparsely reported (30). In *P. aeruginosa*, CAZ-AVI-resistant PDC variants (e.g., $\Delta$D217-Y221 and $\Delta$R210-E219) have been identified (31). The overexpression of $bla_{PDC}$ and an efflux pump and decreased membrane permeability had been reported to be associated with CAZ-AVI resistance (24, 32–34). However, experiments in PAO1 indicated that this issue remains controversial (31). Promoter region variation could also affect gene expression. A previous study in *Enterobacteriaceae* indicated that there were three promoters between IS*Kpn27* and $bla_{KPC-2}$ and that the promoter $P_Y$ is of the utmost importance (35). Since almost all of the $bla_{KPC-2}$ genes are located in the core $\Delta$Tn*6296* platform and the promoter region sequences were identical to the prototype sequence, we argued that promoter variation was unlikely to interfere with $bla_{KPC-2}$ expression in our samples.

There are some limitations of our study. One is that the sampling hospitals are limited and mainly in East China. Therefore, the actual prevalence of KPC-PA across the whole country remains uncertain. As discussed above, our investigation could be seen as a pilot study, and thus, we highly recommend screening for $bla_{KPC}$ genes in nationwide antimicrobial surveillance studies. The other potential limitation is that the Kraken2 taxonomy identifier cannot identify integrative and conjugative elements (ICEs) and prophages that either stay independent (like a plasmid) or integrate onto the chromosome. In addition, we selected contigs whose lengths were >8 kb, which might exclude smaller plasmid contigs. As a consequence, we supposed that strain ZPPH1 harbored type I, VII, and VIII plasmids, but Nanopore sequencing indicated that it harbored two more circularized replicons, a prophage and a 3-kb-long plasmid. We found this small plasmid in 20 strains. However, we believe that this technical limitation does not substantially change the main results of our study.

**Conclusions.** Our study clearly shows that KPC-PA represents a threat to the health care system in China. Therefore, we propose to screen for $bla_{KPC}$ genes in CRPA isolates in nationwide surveillance projects. We believe that such studies are able to guide the therapeutic deployment of CAZ-AVI for the treatment of KPC-PA infections.

## MATERIALS AND METHODS

Additional details are provided in the supplemental material at https://doi.org/10.6084/m9.figshare.14802648.v6.

**Sample collection and antimicrobial susceptibility tests.** Clinical CRPA isolates were collected from seven hospitals around China, including Sir Run Run Shaw Hospital (SRRSH), the First Affiliated Hospital of Zhejiang University (FAHZU), the Provincial People's Hospital of Zhejiang (ZPPH), Quzhou People's Hospital (QZPH), Nanjing Drum Tower Hospital (NDTH), Wuhan Tongji Hospital (WTJH), and the Peking Union Medical College Hospital (PUMCH). The $bla_{KPC}$ gene was screened by PCR with primers KPC-2_FW and KPC-2_RV (see Table S4 in the supplemental material).

*In vitro* antibiotic susceptibilities were determined by broth microdilution or agar dilution methods. Breakpoints were determined according to Clinical and Laboratory Standards Institute (CLSI) document M100, 30th ed. (36). The definition of CRPA is a *P. aeruginosa* strain that exhibits MICs of either imipenem or meropenem of ≥8 mg/liter. *P. aeruginosa* strain ATCC 27853 and *K. pneumoniae* strain ATCC 700603 were used as the quality controls.

**Whole-genome sequencing and *de novo* assembly.** For 151 KPC-PA strains, genomic DNA was extracted using a QIAamp DNA minikit (Qiagen, Hilden, Germany) according to the manufacturer's instructions. Libraries were prepared using the TruePrep DNA library prep kit V2 for Illumina (Vazyme Biotech, Nanjing, China). Sequencing was performed on an Illumina X Ten platform (Illumina Inc., CA, USA). The 150-bp paired-end reads were generated and *de novo* assembled using shovill v1.1.0 (37) with the options "–mincov 10 –minlen 200 –trim." SPAdes v.3.13-v.3.14 (38) was used for assembly.

***Pseudomonas* plasmid sequence identification.** The Illumina short reads were mapped onto three representative plasmid sequences (GenBank accession numbers KY296095.1, MN433457.1, and KU578314.1) using bwa-mem v.0.7.17 (39). The plasmids were categorized into three types based on

sequencing coverage (with a 50% cutoff). A Kraken2-based method as described in a recent study (40) was implemented to identify plasmid contigs. A customized Kraken2 library was built using 233 complete chromosomes and 97 circular *Pseudomonas aeruginosa* plasmids from the *Pseudomonas* Genome Database (version 20.2) and the NCBI database, respectively (Table S5). Contigs that potentially belonged to pseudomonal plasmids were extracted using the Kraken2 taxonomy identifier (41, 42). Contigs belonging to two main plasmid types (types I and II) were filtered out by mummer (43). The criterion was a ratio of the matched length to the contig length of >50%. The remaining contigs, longer than 8 kb and potentially other uncharacterized plasmids, were clustered by CD-HIT-EST v.4.8.1 with the options "-c 0.9 -A 0.9" (44), and their plasmid derivation was obtained by BLAST analysis (45) of the NCBI database. The plasmid composition of each strain, defined as the plasmid pattern, was preliminarily inferred.

**Nanopore long-read sequencing and hybrid assembly.** Representative strains of each plasmid pattern ($n$ = 22) were selected for Nanopore MinION long-read sequencing (Oxford Nanopore Technologies, Oxford, UK). Nanopore long reads of each sample were hybrid assembled with the corresponding Illumina short reads via Unicycler v.0.4.8 (46) or first assembled by canu v.2.0 (47) and further polished using Illumina short reads.

**Sequencing depth measurement.** To estimate the mean sequencing depth of each strain, Illumina short reads were realigned to draft genomes using bwa-mem v.0.7.17 (39). To assess the chromosome sequencing depth, single-copy gene regions were selected to calculate their average depth using samtools v.1.11 (48) with the option "depth -aa." The $bla_{KPC-2}$ gene copy number was represented by the ratio of the sequencing depth of the gene to that of the chromosome.

**Phylogenetic tree construction.** The maximum likelihood phylogenetic tree was generated by RAxML-NG v.1.0.1 (49) with 100 tree searches (50 random and 50 parsimony-based starting trees) and bootstrap replicates (autoMRE criterion). A circular tree layout with an associated heat map was constructed by the R package ggtree v.2.2.4 (50).

**Reverse transcription-quantitative PCR.** Relative KPC expression was measured in 10 strains at the mid-exponential phase (optical density at 600 nm [$OD_{600}$] of ~0.4 to 0.8). Total RNAs were extracted using E.Z.N.A. total RNA kit I (Omega Bio-Tek, GA, USA). Reverse transcription and quantitative PCR (qPCR) were performed using the PrimeScript RT reagent kit with gDNA Eraser and TB green premix ExTaq (TaKaRa, Beijing, China). Specific qPCR primers were designed (Table S4).

**KPC-encoding plasmid curing.** A CRISPR/Cas9-based plasmid-curing method was used to batch cure plasmids. A single-plasmid tool, pCasCurePA, was developed from a previously reported pseudomonal CRISPR/Cas9 gene-editing two-plasmid system (51). The spacer sequence (GCCACGGACCCGTGCAAGCA) was designed in the replicate protein *repA* gene region using sgRNAcas9 v3.0.5 (52). The CRISPR/Cas plasmid curing system was induced by L-arabinose according to a previous study (53).

**Statistical analysis.** All statistical analyses were performed using built-in methods (shapiro.test, dunn.test, fisher.test, cor.test, and p.adjust) in R v.4.0.0-v.4.0.2 and Rstudio v.1.2.5001.

**Ethics approval and consent to participate.** Approval was obtained from the Ethics Committee of Sir Run Run Shaw Hospital (approval/reference number 20201118-49). This study was not considered a human research study. Therefore, no informed consent to participate was required. This study conformed to the principles of the Declaration of Helsinki.

**Data availability.** All the sequence data were deposited in the DDBJ/ENA/GenBank database under BioProject accession number PRJNA672835. The genome sequences of strains NDTH10366 and NDTH9845 have been submitted to GenBank under the following accession numbers: CP064401 (NDTH10366 chromosome), CP064402 (NDTH10366 plasmid), CP073080 (NDTH9845 chromosome), and CP073081 (NDTH9845 plasmid).

## SUPPLEMENTAL MATERIAL

Supplemental material is available online only.

**TABLE S1**, DOCX file, 0.01 MB.
**TABLE S2**, XLSX file, 0.1 MB.
**TABLE S3**, DOCX file, 0.02 MB.
**TABLE S4**, DOCX file, 0.01 MB.
**TABLE S5**, DOCX file, 0.01 MB.

## ACKNOWLEDGMENTS

We thank Quanjiang Ji of ShanghaiTech University for kindly giving us the CRISPR/Cas9 gene-editing tools as a courtesy.

Y.Y. conceived, designed, and coordinated this study. H.S., Z.C., Q.Y., J.Z., X.L., Q.Y., and F.Z. collected the isolates from the respective hospitals. Y.Z. and J.C. performed the microbiological cultures of the isolates, antimicrobial susceptibility tests, plasmid curing, and gene expression measurement. J.J., H.C., Y.L., and L.Z. provided help to extract genomic DNA and perform genome sequencing. Y.Z. and X.H. analyzed the genome sequencing data. Y.Z. wrote the initial version of the manuscript. X.H., S.L., and Y.Y. revised the manuscript. All authors read and approved the final manuscript.

This study is supported by the National Natural Science Foundation of China (grant number 81830069).

We declare that we have no competing interests.

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
