## [Reviewer comments · mSystems]

Deciphering plasmidome of KPC-producing *Pseudomonas aeruginosa* reveals the *bla*_{KPC-2} copy number correlates with ceftazidime-avibactam resistance

Yiwei Zhu, Jie Chen, Han Shen, zhongju chen, Qi-Wen Yang, Jin Zhu, Xi Li, Qing Yang, Feng Zhao, Jingshu Ji, Heng Cai, Yue Li, Linghong Zhang, Sebastian Leptihn, Xiaoting Hua, and Yunsong Yu

Corresponding Author(s): Yunsong Yu, Sir Run Run Shaw Hospital, College of Medicine, Zhejiang University

Review Timeline:

Submission Date:	June 22, 2021
Editorial Decision:	August 12, 2021
Revision Received:	September 15, 2021
Accepted:	October 6, 2021

Editor: Charles Langelier

Reviewer(s): The reviewers have opted to remain anonymous.

Transaction Report:

DOI: <https://doi.org/10.1128/mSystems.00787-21>

August 12, 2021

Prof. Yunsong Yu
Sir Run Run Shaw Hospital, College of Medicine, Zhejiang University
Department of Infectious Diseases
3#, Qingchun east road
Hangzhou, Zhejiang 310016
China

Re: mSystems00787-21 (Deciphering plasmidome of KPC-producing *Pseudomonas aeruginosa* reveals the *bla*_{KPC-2} copy number correlates with ceftazidime-avibactam resistance)

Dear Prof. Yunsong Yu:

Thank you for submitting your manuscript to mSystems. We have completed our review and I am pleased to inform you that, in principle, we expect to accept it for publication in mSystems. However, acceptance will not be final until you have adequately addressed the additional reviewer comments.

Preparing Revision Guidelines

For complete guidelines on revision requirements for your article type, please see the journal Article Types requirement at <https://journals.asm.org/journal/mSystems/article-types>. **Submissions of a paper that does not conform to mSystems guidelines will delay acceptance of your manuscript.**

Sincerely,

Charles Langelier

Editor, mSystems

Journals Department
Reviewer comments:

Reviewer #1 (Comments for the Author):

This resubmitted manuscript addressed most of my previous comments. I like the newly added plasmid curing results, although it could be more convincing if additional blaKPC gene knockout experiments can be conducted in selected isolates, which will directly prove the causal effect of increased blaKPC copy number and CAZ-AVI resistance.

I personally like the original title more. The current title in fact reduced the importance of this study. In my opinion, the study has at least two significant findings. One is the high prevalence of CAZ-AVI resistant KPC-PA in China, and the second one is the molecular mechanism of CAZ-AVI resistance is due to the increased blaKPC copy numbers in different plasmid backgrounds. The authors may consider "Emergence of ceftazidime-avibactam resistant KPC-producing *Pseudomonas aeruginosa* in China" or "High prevalence of ceftazidime-avibactam resistant KPC-producing *Pseudomonas aeruginosa* in China" for the new title. My major comment is that the plasmid grouping method is confusing. The Kraken2 method can only inform whether the contigs were likely from plasmid or chromosome, but doesn't provide grouping information. In addition, customized plasmid and chromosome libraries are needed for the Kraken2 analysis. Which plasmids and chromosomes were used for the analysis? Additional details of the Kraken2 analysis should be provided. It seems that the plasmid grouping was inferred by CD-HIT-EST (line 179). As such, the threshold for clustering should be provided. Was the CD-HIT clustering done by individual blaKPC contigs (line 375) or all concatenated plasmid contigs from the same strain? Were the type I to VIII assigned by CD-HIT? or a hybrid approach involving both reads mapping and CD-HIT (line 356-392)?

What are the definitions/differences of plasmid type (e.g. line 375), pattern (line 377) and group (line 378)? It doesn't seem to be practical to assign all plasmids simply based on the short-reads assemblies. The comprehensive plasmid grouping for each isolate may not add much value to the study. I would suggest the authors simplify the analysis.

Based on the mapping or blast analysis, the genomes with high identities and query coverages against the reference plasmids may be assigned to the corresponding plasmid type (type I, type II and unclassified?), as the analysis confirms the presence of the reference plasmid backbones. Selected strains with type I, II and unclassified plasmid types can be further characterized by nanopore sequencing, in order to probe the molecular mechanisms underlying the increased blaKPC copy numbers.

The results of the study seem interesting. However, the current manuscript generally lacks focus, and is hard to follow. The method section is lengthy. The method section can be trimmed and extra details can be moved to suppl data. Numerous grammar issues were found (e.g. line 64, 100-102, 324 etc) through the manuscript. All these elements should be improved in the resubmission.

Minor comments

Abstract: Line 54-55: better to say "blaKPC plasmid curing in xx strains re-sensitize the CAZ-AVI susceptibility, suggesting xxx". Molecular mechanisms of increased blaKPC copy numbers should be added in the abstract.

Line 106-109: A very brief introduction of the three plasmids may be added.

Line 118: I believed a 50.3% resistance rate should be considered as "High".

Line 152: Specify how many strains were subject to WGS.

Line 166-188: See my above comments about plasmid analysis.

Line 172-176: As I commented above, Kraken2 based method can not assign the contigs in specific plasmids.

Line 179-181: The statement is not accurate. I don't think this approach can determine the "exact" content, and additional details on how the plasmid "patterns" were classified should be provided.

Line 185-186: Specify how many strains were sequenced by nanopore.

Line 205: Why don't use more than one plasmid gene as references (similar as the single copy chromosome genes)?

Line 250: Specify how many samples were characterized by qRT-PCR.

Line 278-279: The space sequences should be provided (in the suppl data).

Line 343-346: Any alternative explanations?

Line 404-408: Add examples on how the transposition and duplication caused blaKPC copy number changes.

Line 410-420: Less relevant. May be removed.

Line 421-432: Not relevant. Can be removed.

Tab S3-1, -2: Similar sample sources and departments/wards may be combined.

Response to Reviewer Comments

We thank reviewers for all their invaluable comments. Our responses are shown below and modifications are highlighted in the revised manuscript.

Reviewer #1:

This resubmitted manuscript addressed most of my previous comments. I like the newly added plasmid curing results, although it could be more convincing if additional blaKPC gene knockout experiments can be conducted in selected isolates, which will directly prove the causal effect of increased blaKPC copy number and CAZ-AVI resistance.

I personally like the original title more. The current title in fact reduced the importance of this study. In my opinion, the study has at least two significant findings. One is the high prevalence of CAZ-AVI resistant KPC-PA in China, and the second one is the molecular mechanism of CAZ-AVI resistance is due to the increased blaKPC copy numbers in different plasmid backgrounds. The authors may consider "Emergence of ceftazidime-avibactam resistant KPC-producing *Pseudomonas aeruginosa* in China" or "High prevalence of ceftazidime-avibactam resistant KPC-producing *Pseudomonas aeruginosa* in China" for the new title.

Response: We are pleased to learn that the reviewer likes our study. We agree that *bla*_{KPC} gene knock-out would be more convincing. Therefore, we have knocked out the single-copy *bla*_{KPC-2} gene on the plasmid in strain SRRSH1408. Both of the SRRSH1408 strains, *bla*_{KPC-2}-KO and plasmid-cured, shared the same susceptibility to CAZ-AVI, 4mg/L. This indicates that the plasmid-curing approach has a comparable effect with the gene knock-out approach. Knocking out multiple-copy *bla*_{KPC-2} genes was not successful despite several attempts. We hope the reviewer understands that we would like to publish our study at this stage. We also thank the reviewer for his recommendation of titles for our paper and gratefully chose the first one "Emergence of ceftazidime-avibactam resistant KPC-producing *Pseudomonas aeruginosa* in China" as the title of our revised paper.

Major comment:

1. My major comment is that the plasmid grouping method is confusing. The Kraken2 method can only inform whether the contigs were likely from plasmid or chromosome, but doesn't provide grouping information. In addition, customized plasmid and chromosome libraries are needed for the Kraken2 analysis. Which plasmids and

chromosomes were used for the analysis? Additional details of the Kraken2 analysis should be provided. It seems that the plasmid grouping was inferred by CD-HIT-EST (line 179). As such, the threshold for clustering should be provided. Was the CD-HIT clustering done by individual blaKPC contigs (line 375) or all concatenated plasmid contigs from the same strain? Were the type I to VIII assigned by CD-HIT? or a hybrid approach involving both reads mapping and CD-HIT (line 356-392)?

Response: We implemented the Kraken2 method to extract plasmid contigs. To construct a customized Kraken2 library, we downloaded 233 complete chromosomes and 97 circular *Pseudomonas aeruginosa* plasmids from Pseudomonas Genome Database (version 20.2) and NCBI database, respectively. All GenBank accession numbers of these sequences are provided in the Supp Table S5. For plasmid contigs, we further filtered out contigs belonging to two main plasmid types (Type I and II) by mummer. The criterium is the ratio of matched length to the contig length greater than 50%. The remaining contigs potentially belong to other uncharacterized plasmids. We extracted unclassified plasmid contigs longer than 8 kb (n=74). We classified these 74 contigs into 22 clusters by CD-HIT-EST (options: -c 0.9 -A 0.9). Representative contig from each cluster were aligned in GenBank database by BLASTN.

2. What are the definitions/differences of plasmid type (e.g. line 375), pattern (line 377) and group (line 378)? It doesn't seem to be practical to assign all plasmids simply based on the short-reads assemblies. The comprehensive plasmid grouping for each isolate may not add much value to the study. I would suggest the authors simplify the analysis.

Response: We define plasmid type characterized by the replication or transfer system. Plasmid pattern refers to the composition of plasmids in a single strain. We have now corrected the word "group" and refer to it as plasmid pattern. We have followed the reviewer's comment to simplify the analysis.

3. Based on the mapping or blast analysis, the genomes with high identities and query coverages against the reference plasmids may be assigned to the corresponding plasmid type (type I, type II and unclassified?), as the analysis confirms the presence of the reference plasmid backbones. Selected strains with type I, II and unclassified plasmid types can be further characterized by Nanopore sequencing, in order to probe the molecular mechanisms underlying the increased blaKPC copy numbers.

Response: We agree with the reviewer. We first identified the presence of Type I and II

plasmids by mapping analysis and found further plasmids by contig blast analysis. We selected representative strains from each plasmid pattern for Nanopore sequencing, which generally confirmed our plasmid cluster analysis of the NGS data.

4. The results of the study seem interesting. However, the current manuscript generally lacks focus, and is hard to follow. The method section is lengthy. The method section can be trimmed and extra details can be moved to suppl data. Numerous grammar issues were found (e.g line 64, 100-102, 324 etc) through the manuscript. All these elements should be improved in the resubmission.

Response: We thank the reviewer's suggestion. The method section has been simplified. We have moved details of methods to the supplemental information. Grammar issues are corrected (Line 64, 101, 224-226) in the revised manuscript.

Minor comments

1. Abstract: Line 54-55: better to say "blaKPC plasmid curing in xx strains re-sensitize the CAZ-AVI susceptibility, suggesting xxx ". Molecular mechanisms of increased blaKPC copy numbers should be added in the abstract.

Response: We thank the reviewer for his/her suggestion and have modified this sentence in the revised manuscript (Line 53-54).

2. Line 106-109: A very brief introduction of the three plasmids may be added.

Response: We have added a brief introduction of the three plasmids in the revised manuscript (Line 105-109)

3. Line 118: I believed a 50.3% resistance rate should be considered as "High".

Response: We appreciated the reviewer's comment, and have corrected it (line 118).

4. Line 152: Specify how many strains were subject to WGS.

Response: 151 strains are subjected to WGS (seen Line 142).

5. Line 166-188: See my above comments about plasmid analysis.

Response: Please see our response above (major comment 1).

6. Line 172-176: As I commented above, Kraken2 based method cannot assign the contigs in specific plasmids.

Response: We agreed with the reviewer that it cannot assign the contigs into specific plasmids. The Kraken2-based contigs clustering is the first step of plasmid composition analysis. It should be further confirmed by long-read sequencing. We have corrected this statement in the revised manuscript (line 153-154).

7. Line 179-181: The statement is not accurate. I don't think this approach can determine the "exact" content, and additional details on how the plasmid "patterns" were classified should be provided.

Response: We agreed with the reviewer this approach does have limitations as we pointed out in the discussion (line 428-438). The plasmid patterns classification should be further confirmed by long-read sequencing. We have corrected this statement as a preliminary classification in the revised manuscript (line 167-168).

8. Line 185-186: Specify how many strains were sequenced by nanopore.

Response: 22 strains were sequenced by Nanopore. We have added this information in Line 170 in the revised manuscript.

9. Line 205: Why don't use more than one plasmid gene as references (similar as the single copy chromosome genes)?

Response: We agree with the reviewer that it is more reasonable to use more than one plasmid gene as reference. Since we have removed the plasmid copy number analysis in the revised manuscript and only kept the analysis of blaKPC-2 gene copy number, its expression and CAZ-AVI MIC, it becomes meaningless to determine the plasmid copy number. Therefore, we have removed this sentence in the Methods & Materials section.

10. Line 250: Specify how many samples were characterized by qRT-PCR.

Response: Ten samples were characterized by qRT-PCR (line 190).

11. Line 278-279: The spacer sequences should be provided (in the suppl data).

Response: The spacer sequence is now provided in revised manuscript (line 201).

12. Line 343-346: Any alternative explanations?

Response: Alternative explanation might be clonal transmission of KPC-PA strains among patients with urethral catheterization. However, this kind of data is lacking in our study. We believe that investigating the underlying mechanism that causes the high percentage of KPC-PA in urine is beyond the scope of this study. In the revised manuscript, we remove this unsubstantiated statement. After combining similar sample sources as required in the Minor comment 16, KPC-PA was significantly associated with urine and abdominal drainage (Supp Table S3). We comment that it endorses the use of CAZ/AVI to treat infections at these two anatomical sites. (line 244-247)

13. Line 404-408: Add examples on how the transposition and duplication caused blaKPC copy number changes.

Response: We have added examples in the revised manuscript (Line 289-290).

14. Line 410-420: Less relevant. May be removed.

Response: We have removed these sentences.

15. Line 421-432: Not relevant. Can be removed.

Response: We have removed these sentences.

16. Tab S3-1, -2: Similar sample sources and departments/wards may be combined.

Response: We have combined similar sample sources and departments/wards in the revised Supplemental table S3.

October 6, 2021

Prof. Yunsong Yu
Sir Run Run Shaw Hospital, College of Medicine, Zhejiang University
Department of Infectious Diseases
3#, Qingchun east road
Hangzhou, Zhejiang 310016
China

Re: mSystems00787-21R1 (Deciphering plasmidome of KPC-producing *Pseudomonas aeruginosa* reveals the *bla*_{KPC-2} copy number correlates with ceftazidime-avibactam resistance)

Dear Prof. Yunsong Yu:

I'm please to inform that your manuscript has been accepted, and I am forwarding it to the ASM Journals Department for publication. For your reference, ASM Journals' address is given below. Before it can be scheduled for publication, your manuscript will be checked by the mSystems senior production editor, Ellie Ghatineh, to make sure that all elements meet the technical requirements for publication. She will contact you if anything needs to be revised before copyediting and production can begin. Otherwise, you will be notified when your proofs are ready to be viewed.

As an open-access publication, mSystems receives no financial support from paid subscriptions and depends on authors' prompt payment of publication fees as soon as their articles are accepted. =

Publication Fees:

We recognize that the video files can become quite large, and so to avoid quality loss ASM suggests sending the video file via <https://www.wetransfer.com/>. When you have a final version of the video and the still ready to share, please send it to Ellie Ghatineh at eghatineh@asmusa.org.

Sincerely,

Charles Langelier
Editor, mSystems

Journals Department
Supplemental Table S1: Accept
Supplemental Table S3: Accept
Supplemental Table S5: Accept
Supplemental Table S2: Accept
Supplemental Table S4: Accept